# The Clinical Utility of the Geriatric Nutritional Risk Index in Predicting Postoperative Complications and Long-Term Survival in Elderly Patients with Colorectal Cancer after Curative Surgery

**DOI:** 10.3390/cancers13225852

**Published:** 2021-11-22

**Authors:** Chun-Kai Liao, Yih-Jong Chern, Yu-Jen Hsu, Yueh-Chen Lin, Yen-Lin Yu, Jy-Ming Chiang, Chien-Yuh Yeh, Jeng-Fu You

**Affiliations:** 1Division of Colon and Rectal Surgery, Department of Surgery, Chang Gung Memorial Hospital, Linkou, Taoyuan 333423, Taiwan; MR9023@cgmh.org.tw (C.-K.L.); b9202063@cgmh.org.tw (Y.-J.C.); m8295@cgmh.org.tw (Y.-J.H.); ld@cgmh.org.tw (Y.-C.L.); jmjiang@adm.cgmh.org.tw (J.-M.C.); chnyuh@gmail.com (C.-Y.Y.); 2Division of Colon and Rectal Surgery, Department of Surgery, Chang Gung Memorial Hospital, Keelung Branch, Keelung City 204201, Taiwan; tomyuauk@cgmh.org.tw; 3School of Medicine, Chang Gung University, Taoyuan 333323, Taiwan

**Keywords:** geriatric nutritional risk index (GNRI), colorectal cancer, complication, prognosis

## Abstract

**Simple Summary:**

Elderly cancer patients usually suffer with malnutrition. Preoperative malnutrition has been considered a poorer prognostic factor in cancer treatment. The geriatric nutritional risk index (GNRI) is a simple tool for predicting the risk of morbidity and mortality in elderly patients by using albumin, height, and body weight parameters. In this study, we evaluated whether GNRI is a reliable marker for postoperative complications and long-term survival. By retrospectively evaluating 1206 CRC patients aged over 75 years who underwent curative-intent surgery at Chang Gung Memorial Hospital, there were significantly more postoperative complications in the low GNRI group (*p* < 0.001) and GNRI was an independent risk factor for postoperative complications (HR: 1.774, *p* = 0.037). Overall survival and disease-free survival were significantly worse in the low GNRI group (both *p* < 0.001) and a GNRI < 98 was statistically identified as an independent prognostic factor for survival. Conclusively, GNRI can be a reliable biomarker in clinical practice.

**Abstract:**

Research on the relationship between the geriatric nutritional risk index (GNRI) and postoperative complications/oncological outcomes in elderly colorectal cancer (CRC) patients is limited. This study investigated the prognostic value of the GNRI in aged CRC patients. We retrospectively analyzed 1206 consecutive CRC patients aged over 75 years who underwent curative-intent surgery from January 2008 to December 2015 and categorized them into high GNRI (≥98) and low GNRI (<98) groups according to a receiver operating characteristic (ROC) curve analysis. Uni- and multivariate logistic regression analysis were used to explore the association of the GNRI with postoperative complications. Kaplan–Meier survival analyses and the Cox proportional hazard model were used to explore the association between GNRI and survival. We discovered that GNRI is an independent risk factor for postoperative complications (HR: 1.774, *p* = 0.037). Surgical site infection, wound dehiscence and pneumonia were more common in patients with GNRI < 98. Survival analysis showed significantly worse overall survival and disease-free survival in the low GNRI group (both *p* < 0.001). In the multivariate analysis, GNRI < 98 was an independent risk factor for OS (HR: 1.329, *p* = 0.031) and DFS (HR: 1.312, *p* = 0.034). Thus, preoperative GNRI can be effectively used to predict postoperative complications and long-term survival in elderly CRC patients after curative surgery.

## 1. Introduction

Colorectal cancer (CRC) is the third leading malignancy and the second highest cause of cancer-related death worldwide. According to GLOBOCAN statistics, there were more than 1.9 million new CRC cases and 935,000 related deaths estimated in 2020 [1]. In the United States, an estimated 54% of new CRC cases in 2020 occurred in patients over 65 years of age [2]. Elderly cancer patients usually have several comorbidities [3] accompanied by malnutrition [4,5,6]. Several studies have revealed that pretreatment malnutrition status is a risk factor for posttreatment complications and worse oncologic outcomes [7,8]. Thus, a validated tool for examining nutritional status is important for risk stratification and for determining the optimal treatments for elderly patients.

The geriatric nutritional risk index (GNRI) was first described by Bouillanne et al. [9] and is a simple tool for predicting the risk of morbidity and mortality in elderly patients by using albumin and body weight parameters. Previous studies have shown that a lower GNRI can predict posttreatment complications, longer hospital stays and long-term mortality in elderly patients [10,11,12]. Recently, many studies have revealed the prognostic role of the GNRI in a variety of cancers, including gastric cancer, esophageal cancer, hepatocellular carcinoma, and lung cancer [13,14,15,16]. Nevertheless, few studies about the GNRI and its relationship to CRC have been reported, and the clinical value is limited because only a small cohort of patients have been assessed [17,18]. Additionally, previous studies mostly defined patients aged over 65 years as the elderly group. This is not an accurate classification for elderly patients because most patients in their mid-sixties are not as fragile as those in their mid-seventies in clinical practice. Thus, this study aimed to investigate the prognostic value of the GNRI in postoperative complications and the long-term outcomes for elderly CRC patients after curative surgery, focusing especially on elderly patients over 75 years old.

## 2. Materials and Methods

### 2.1. Patient Selection

We retrospectively analyzed 1206 consecutive patients aged over 75 years who were diagnosed with CRC and underwent surgery at Chang Gung Memorial Hospital between January 2008 and December 2015. The inclusion criteria were set as follows: (1) age ≥ 75 years; (2) received curative-intent radical surgery; (3) pathologically confirmed CRC origin; (4) no metastatic lesions; and (5) complete preoperative data and clinicopathological characteristics. Patients who did not fulfill the inclusion criteria were excluded from the analysis. The flow chart of clinical case selection is shown in Figure 1. The Institutional Review Board of Chang Gung Memorial Hospital approved this study with approval number 202101119B0. All used data were recorded in the hospital database and used for research purposes only.

### 2.2. Data Collection

We retrieved and retrospectively analyzed data from the Colorectal Section Tumor Registry in Chang Gung Memorial Hospital. This database is a prospectively designed database consisting of the records of postoperative patients who were consecutively and actively followed up.

Preoperative variables, including age, sex, comorbidities, body mass index (BMI), preoperative serum albumin level, white blood cell (WBC) count, hemoglobin (Hb), and carcinoembryonic antigen (CEA), were analyzed. Blood samples were obtained 1 week prior to the surgery. The Charlson comorbidity index (CCI) score was used to categorize the comorbidities of the patients. CCI scores of 1 to 2 were categorized as mild grade; CCI scores of 3 to 4 were categorized as moderate grade; and CCI scores of 5 or more were categorized as severe grade. The clinicopathological parameters, including operative type (emergency or elective), operative method (laparoscopy or open), tumor location, T stage, N stage, histological grade, and histological type, were also analyzed. The stage was classified according to the eighth edition of the Union for International Cancer Control (UICC) tumor-node-metastasis (TNM) classification. The postoperative outcomes, including postoperative complications, mortality, overall survival (OS) and disease-free survival (DFS), were assessed. Postoperative complications were classified according to the Clavien-Dindo (CD) grade.

### 2.3. Nutritional Assessment by GNRI and Other Parameters

The formula of GNRI was GNRI = 1.487 × serum albumin concentrations (g/L) + 41.7 × preoperative body weight (PBW)/ideal body weight (IBW) [9]. In this study, we defined the ideal body weight as IBW = height ^2^ (m) × 22. When the PBW of patients exceeded the IBW, the PBW/IBW was set at 1. The other parameters used in this study included the prognostic nutrition index (PNI) and the neutrophil–lymphocyte ratio (NLR). The formulas for PNI and NLR were as follows: PNI = (10 × serum albumin [g/dL]) + (0.005 × lymphocytes/μL); NLR = absolute neutrophil count/absolute lymphocyte count.

### 2.4. Follow-Up

All physicians adopted similar follow-up routines and adjuvant treatment protocols in this institution. After primary tumor resection, all patients were subjected to a follow-up program that included outpatient visits every 3 to 6 months for physical examinations and CEA tests. Chest radiography, abdominal ultrasonography, or abdominal computed tomography (CT) imaging, in addition to colonoscopy, were performed one year after surgery and every 1 to 2 years whenever necessary. Follow-up status was recorded postoperatively every 12 months by a team of five specially trained nurses and validated by two physicians. The date of first recurrence was defined as the first date when the existence of local recurrence and/or distant metastases was confirmed by histology of biopsy specimens, additional surgery, and/or radiological studies. The index date for survival calculation was the date of surgery for primary cancer. OS was defined as the interval from cancer resection to death or last follow-up. DFS was defined as the interval from cancer resection to the date of first recurrence, death, or last follow-up. The last follow-up date in this study was 31 July 2020.

### 2.5. Statistical Analysis

Pearson’s chi-squared test or Fisher’s exact test were used to compare the categorical variables, whereas Student’s t test was used for continuous variables. The survival analysis was performed using Kaplan–Meier curves with the log-rank test. Univariate and multivariate analyses were performed using the Cox proportional hazards model to assess the risk factors associated with OS and DFS. Univariate and multivariate logistic regression analyses were applied to assess the factors associated with postoperative complications. A receiver operating characteristic (ROC) curve analysis was used to identify the optimal cutoff values of GNRI, NLR, and PNI for OS evaluation. Differences with a two-sided *p* < 0.05 were considered statistically significant. All parameters were analyzed using the Statistical Package for Social Sciences (SPSS) version 24 (IBM Corp., New York, NY, USA) and GraphPad Prism version 9 (GraphPad Software Inc., San Diego, CA, USA).

## 3. Results

### 3.1. Patient Characteristics

Our study enrolled 1206 patients who underwent a radical resection for CRC between 2008 and 2015. Of these patients, 673 were males and 573 were females. The mean age was 80.45 ± 4.42 years. The average BMI was 23.45 ± 3.70, with a range from 13.22 to 39.96. Four hundred twenty-three (35.1%) patients were categorized as mild grade (CCI score 1–2), 627 (52%) patients were categorized as moderate grade, (CCI score 3–4), and 156 (12.9%) patients were categorized as severe grade (CCI score ≥ 5) of comorbidity. The median follow-up time was 60.67 months, with a range from 1 to 146.5 months. There were 187 (15.5%) patients with stage I disease, 527 (43.7%) patients with stage II disease, and 492 (40.8%) with stage III disease. Two hundred and twenty (18.2%) patients experienced postoperative complications, of whom 145 (12%) patients had CD grade ≥ 2 complications. The detailed characteristics of all patients are listed in Table 1.

### 3.2. The Association of GNRI and Clinicopathological Factors

The GNRI of all patients ranged from 59.96 to 116.05, with a mean value of 96.92 ± 8.82. The ROC curve analysis for OS in this study showed that the optimal cutoff value of the GNRI was 97.98 (area under the curve = 0.643, sensitivity = 0.664, and specificity = 0.569) (Figure 2a). We categorized the patients into a low-risk GNRI (GNRI ≥ 98) group and a high-risk GNRI (GNRI < 98) group according to the ROC analysis. There were 544 (45.1%) and 662 (54.9%) patients in the high- and low-risk groups, respectively. There was no significant difference between the low- and high-risk GNRI groups in terms of sex or number of lymph node metastases. Nonetheless, we found that patients with a GNRI < 98 were older (81.58 ± 4.80 vs. 79.51 ± 3.84 years, *p* < 0.001), had a lower BMI (21.74 ± 3.57 vs. 24.85 ± 3.70, *p* < 0.001), had lower albumin levels (48.5% vs. 0%, *p* < 0.001), were right-sided predominant (36.9% vs. 22.7%, *p* < 0.001), had more advanced tumor invasion (T3/4: 86.4% vs. 75.4%, *p* < 0.001), and had poorer histological grades and histological features. More patients in the high-risk GNRI group had severe CCI scores (19.3% vs. 7.7%, *p* < 0.001). Other associated factors between the two groups were preoperative CEA level, WBC count, Hb, NLR, PNI, operative type, and operative methods. The details of the association between clinicopathological factors and GNRI status are listed in Table 2.

### 3.3. The Association of GNRI and Postoperative Complications

As shown in Figure 2b, the preoperative GNRI was significantly lower in patients with postoperative CD grade ≥ 2 complications than in those without complications. The mean GNRI was 91.81 ± 10.28 in the complication subgroup and 97.62 ± 8.37 in the no-complication subgroup (*p* < 0.001). The postoperative complication rate was significantly higher in the high-risk GNRI group than in the low-risk group (25% vs. 12.7%, *p* < 0.001). The details of the association between GNRI and postoperative complications are listed in Table 3. There was no significant difference between the low- and high-risk GNRI groups in terms of postoperative ileus, anastomosis leakage, intra-abdominal infection, and cardiovascular events. However, more surgical site infections, wound dehiscence, pneumonia, and urinary tract infections occurred in the high-risk GNRI group. There were more CD grade ≥ 2 complications in the high- than in the low-risk GNRI group (18.4% vs. 6.8%, *p* < 0.001). The mortality rate was also higher in the high-risk GNRI group (2.9% vs. 0.3%, *p* < 0.001).

### 3.4. Risk Factors for CD Grade ≥ 2 Complications

The univariate and multivariate logistic regression analysis results are listed in Table 4. In the univariate analysis, age ≥ 85 years (*p* = 0.06), CCI score ≥ 5 (*p* < 0.001), low albumin (*p* < 0.001), high WBC count (*p* = 0.037), low PNI (*p* < 0.001), high NLR (*p* < 0.001), low GNRI (*p* < 0.001), and emergency surgery (*p* = 0.007) were significantly associated with CD grade ≥ 2 complications. In the multivariate analysis, only a CCI score ≥ 5 (HR: 1.601, 95% CI: 1.004–2.54, *p* = 0.048) and a low GNRI (HR: 1.805, 95% CI: 1.057–3.083, *p* = 0.03) were independent risk factors for CD grade ≥ 2 complications.

### 3.5. Survival Analysis and Prognostic Factors for OS

In the survival analysis, patients in the high-risk GNRI group had worse survival rates than those in the low-risk GNRI group (Figure 3). The 1-year, 3-year, and 5-year OS rates were 96.21%, 84.47%, and 73.43% in the low-risk GNRI group and 84.51%, 64.57%, and 47.51% in the high-risk GNRI group, respectively (*p* < 0.001). The univariate analysis showed that age ≥ 85 years, CCI score ≥ 5, male sex, low BMI, low albumin, high WBC, low Hb, high CEA, high NLR, low PNI, low GNRI, T4 tumor grade, high number of lymph node metastases, poor histological features and grade, and emergency surgery were associated with poor OS. The multivariate analysis showed that age ≥ 85 years (HR: 2.046, *p* < 0.001), male sex (HR: 1.332, *p* < 0.001), CCI score ≥ 5 (HR: 2.081, *p* < 0.001), BMI < 22 (HR: 1.236, *p* = *0*.026), low albumin (HR: 1.275, *p* = 0.042), low GNRI (HR: 1.329, *p* = 0.031), and N stage (N1, HR: 1.27, *p* = 0.021; N2, HR: 2.061, *p* < 0.001) were independent prognostic factors for OS. The detailed univariate and multivariate analysis results for OS are listed in Table 5.

### 3.6. Survival Analysis and Prognostic Factors for DFS

In the survival analysis, patients in the high-risk GNRI group had worse DFS than those in the low-risk GNRI group (Figure 4). The 1-year, 3-year, and 5-year OS rates were 90.75%, 78.40%, and 69.32% in the low-risk GNRI group and 77.91%, 58.76%, and 44.23% in the high-risk GNRI group, respectively (*p* < 0.001). The detailed univariate and multivariate analysis results for DFS are listed in Table 6. The multivariate analysis showed that age ≥ 85 years (HR: 1.90, *p* < 0.001), male sex (HR: 1.341, *p* < 0.001), CCI ≥ 5 (HR: 1.938, *p* < 0.001), high CEA (HR: 1.259, *p* = 0.011), low GNRI (HR: 1.312, *p* = 0.034), T4 stage (HR: 2.212, *p* = 0.017), and N stage (N1, HR: 1.345, *p* = 0.003; N2, HR: 2.212, *p* < 0.001) were independent prognostic factors for DFS.

### 3.7. Subgroup Analysis and the Utility of GNRI on Survival

We performed a subgroup survival analysis to examine whether the effect of GNRI on survival changed according to other clinicopathological variables. As shown in Figure 5, patients in the low-risk GNRI group still had a better OS and DFS than those in the high-risk GNRI group, even when stratified by stage I, II, or III. The same effect was observed among patients stratified according to another nutritional marker (PNI < 48 or ≥48) (Figure 6). The details of the subgroup survival analysis are shown in Figure 7. Patients with a high-risk GNRI had an increased risk of worse OS in most subgroups except those with CCI scores ≥ 5, those who received emergency surgery, or those with poor histological features. The same effect of GNRI on DFS also existed in most subgroups except those with CCI scores of ≥ 5, those who received emergency surgery and those who had poor histological features and N2 stage.

## 4. Discussion

Malnutrition is an important issue in treating cancer patients, especially patients with gastrointestinal (GI) cancer. According to a previous study, the prevalence of malnutrition is high and usually underestimated in patients with GI cancers [19]. Preoperative malnutrition can lead to more postoperative complications and infections, longer hospital stays, and worse oncological outcomes [7,8,20,21]. Thus, a validated tool for assessing patients before starting treatment is indispensable for identifying high-risk patients and addressing malnutrition before major treatment. Several nutritional markers have been proposed to be associated with cancer prognosis, including albumin, PNI, BMI, sarcopenia, and many other serum markers [22,23,24,25,26,27,28,29]. However, the cut-off values of these markers differed from study to study, which limits their universal utility in clinical practice.

Currently, several nutritional screening scales such as Malnutrition Universal Screening Tool (Must) or Nutritional Risk Screening 2002 (NRS-2002) have been proposed [30]. However, these scales require the records about weight loss in the past 3–6 months. In clinical practice, elderly patients may not record their body weight frequently and they may usually not remember their usual body weight. For elderly patients with poor performance or bedridden status, the measurement of body weight is more difficult, and the records of BW change are usually lacking. The GNRI is a simple marker first described by Bouillanne et al. in 2005 [9]. The concept of GNRI is to use the ratio of current BW and ideal BW to simulate the BW change. This formula helps identify high risk patients by using parameters such as preoperative body weight, height, and albumin data, which are usually easy to obtain while starting treatment. Because of the setting of present BW/ideal BW as equal to 1 while present BW over the ideal BW, malnourished patients with overweight could be diagnosed. Although this formula leads to a higher weighting for albumin than for weight, Bouillanne et al. indicates that the risk of mortality is lower in obese patients than in low BMI patients.

The GNRI was originally categorized into four risk groups with cutoff values of 82, 92, and 98 [9]. Many studies have been conducted using this classification system and have indicated its prognostic value [15,18]. Recently, many studies have divided the GNRI into two classifications by using the ROC analysis to determine the optimal cutoff values. In a recent meta-analysis conducted by Xie et al. [31] comparing the GNRI groups and the outcome of GI malignancy, the cutoff value of 98 was used for studies regarding esophageal cancer, pancreatic duct cancer, CRC, and CRC with liver metastasis. In this study, the optimal cutoff value was 97.98 according to the ROC analysis, which is compatible with previous studies for risk stratification. According to the above results, a GNRI cutoff of 98 may be universally used for risk stratification.

Previous studies on the GNRI have mostly defined elderly patients as those aged over 65 years. With the advances in medical care and the extension of life spans, most patients in their mid-sixties are still healthy enough to receive surgery or other invasive treatment. Thus, this study enrolled patients older than 75 years for analysis to investigate the value of GNRI in a true “elderly” group. We identified many clinicopathological factors in the high-risk GNRI group. Patients older than 80 years, with multiple comorbidities, had a low BMI, and those with an advanced cancer stage initially tended to present with a low GNRI. Our former study had showed that patients with right-sided colon cancer are more malnourished than left-sided colon cancer, even with the same stage. Moreover, a more advanced tumor invasion and poor pathological features but not lymph node involvement were found in the same cohort [32]. In this study, we demonstrated that more patients with a low GNRI had right-sided origin CRC and advanced staging, which is comparable with our previous study that showed that right-sided colon cancer is accompanied with more malnutrition. Based on above findings, we should pay attention to these poor clinicopathological features when initiating treatment because patients may be at risk of malnutrition and nutrition intervention. In the review of these patients’ preoperative activity, we also noted that more patients with a GNRI ≥ 98 tolerated Chinese Kung-fu/sports ≥ 5 hrs/week (19.9% vs. 12.9%, *p* = 0.001). This finding also suggested that the GNRI can reflect the ability of patients to perform physical activity and indicate their physique.

Duraes el al. [33] showed significantly worse oncologic outcomes and a higher Clavien-Dindo grade in CRC patients who underwent curative resection even with only CD grade 2 complications. Thus, we included all CD grade ≥ 2 events for risk analysis. In this study, patients with a low GNRI had more complications than those with a high GNRI (all grade: 25% vs. 12.7%, *p* < 0.001) and more prominent as CD grade ≥ 2 complications (18.4 vs. 6.8%, *p* < 0.001). In the further analysis of complications, no significant differences were observed in the rates of ileus, anastomosis leakage, or cardiovascular events between the low and high GNRI groups, but a significant increase in the number of cases of surgical site infection, wound dehiscence, pneumonia, and urinary tract infection was found in the low GNRI group. Additionally, we identified that GNRI < 98 was an independent risk factor for CD grade ≥ 2 complications. These findings are compatible with those of another study conducted by Sasaki et al. [17], which enrolled all-stage CRC patients and showed a correlation of GNRI with surgical site infection but not with leakage.

Although previous studies have explored the relationship of GNRI and outcomes in elderly patients with CRC [17,18], they enrolled patients older than 65 years and had small case numbers for analysis. The present study enrolled 1206 patients aged over 75 years and completed follow-up for at least 5 years to assess a true “elderly” cohort. Our findings suggested that a low preoperative GNRI (<98) is associated with poorer OS and DFS than a high preoperative GNRI (≥98) for CRC patients aged ≥ 75 years. The multivariate analysis also indicated that GNRI is a prognostic factor for OS and DFS independent of disease stage and comorbidities. To discover the utility of the GNRI in predicting prognosis, we analyzed the relationship of the GNRI and OS/DFS in the context of different clinicopathological factors. In our study, better OS and DFS were observed in patients with a high GNRI, even in different age, sex, stage, histology, or comorbidity subgroups. Other prognostic markers, such as the NLR or PNI, have been reported to be associated with CRC outcomes [34,35]. This study also showed the same results when the optimal cutoff value for the NLR and PNI obtained from ROC analysis was applied. Patients with a high NLR or low PNI had significantly worse OS and DFS than the opposite group. However, PNI was not an independent factor for OS or DFS in the multivariate analysis. Additionally, patients in different GNRI groups had significantly different survival outcomes even in the low- or high-PNI subgroups or in the low- or high-NLR subgroups. These findings corroborated the rationale and robustness of the cutoff value we applied for the GNRI.

The strength of this study is that, to our knowledge, we assessed the largest cohort in our investigation of the relationship of GNRI and postoperative complications and the prognosis in CRC patients. Moreover, we defined elderly patients as those aged ≥ 75 years, which better represents the most fragile population in the clinical setting. This study is limited by its retrospective nature, and selection bias may exist. The lack of some inflammation markers, such as CRP, prevented us from comparing the association of the GNRI with other inflammation parameters, such as the Glasgow Prognostic Score (GPS) or CRP/albumin ratio (CAR), which are known to be related to nutrition status. Additionally, we had no information on the use universal nutrition screening tools to verify with GNRI and no data about preoperative nutritional intervention, which might have affected the results. Further prospective studies are warranted to answer these questions.

## 5. Conclusions

Our study revealed that low preoperative GNRI (<98) is an independent risk factor for postoperative complications and also causes poor OS and DFS in CRC patients over 75 years old after curative surgery. GNRI can be a reliable biomarker in clinical practice. It is a predictor of prognosis and can be used to further screen high-risk patients from those identified with the existing classification system.

## Figures and Tables

**Figure 1 cancers-13-05852-f001:**
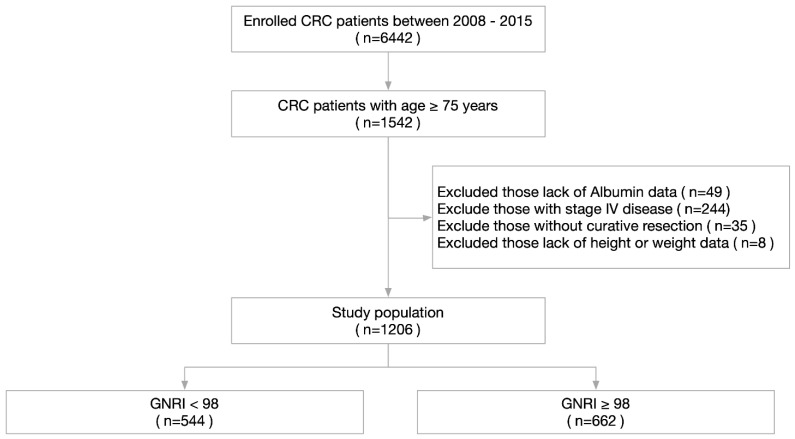
The flowchart illustrating the clinical patient selection in this study.

**Figure 2 cancers-13-05852-f002:**
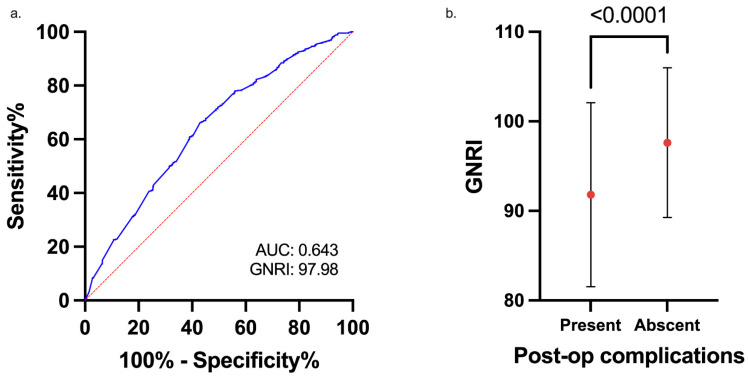
(**a**). Receiver operating characteristic (ROC) curve analysis of the geriatric nutritional risk index (GNRI) for overall survival (OS) in elderly patients with colorectal cancer (CRC). (**b**). The mean GNRI in patients with or without Clavien-Dindo grade ≥ 2 postoperative complications.

**Figure 3 cancers-13-05852-f003:**
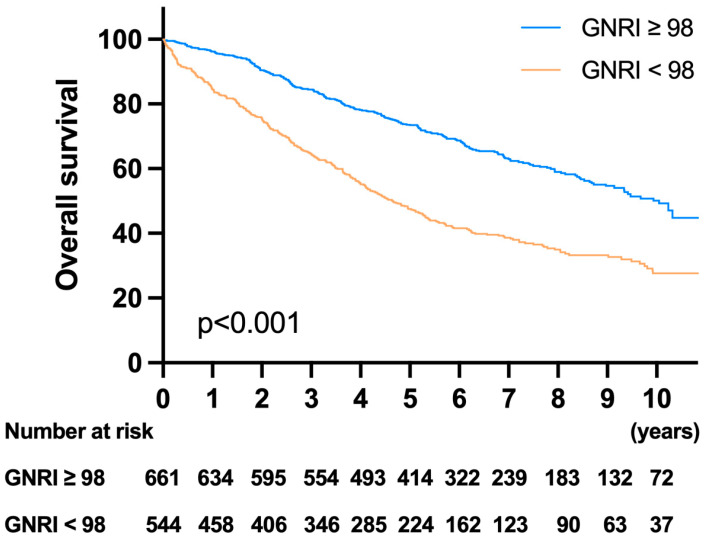
Kaplan–Meier survival analysis of OS according to the GNRI. Patients with a GNRI < 98 had significantly worse survival rates than those with a GNRI ≥ 98.

**Figure 4 cancers-13-05852-f004:**
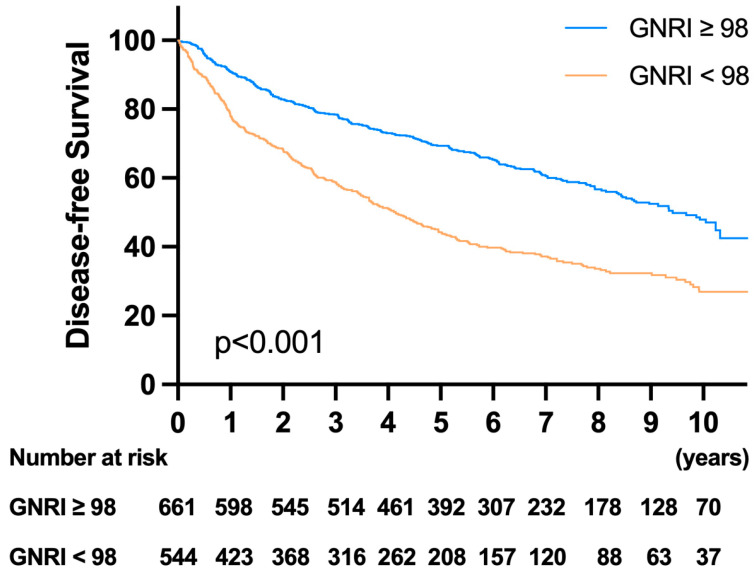
Kaplan–Meier survival analysis of disease-free survival (DFS) according to the GNRI. Patients with GNRI < 98 had significantly worse survival than those with GNRI ≥ 98.

**Figure 5 cancers-13-05852-f005:**
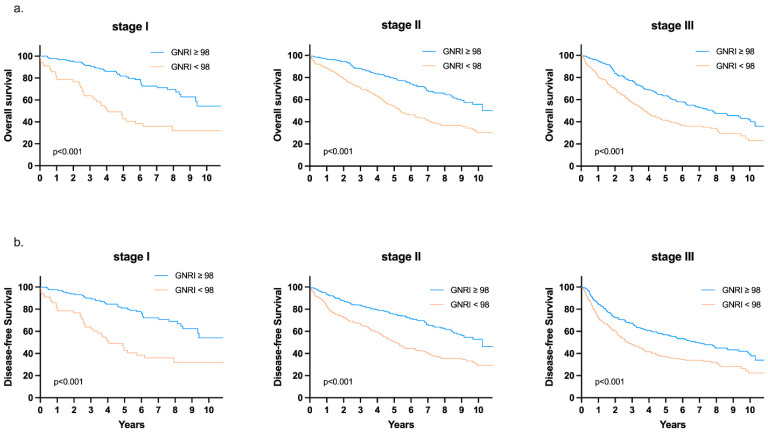
Kaplan–Meier survival analysis of (**a**) OS and (**b**) DFS according to GNRI in stage I, II, and III patients.

**Figure 6 cancers-13-05852-f006:**
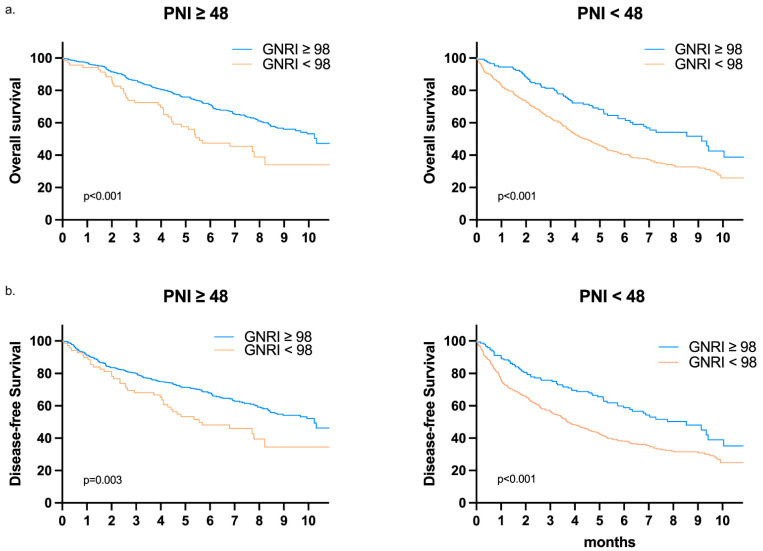
Kaplan–Meier survival analysis of (**a**) OS and (**b**) DFS according to GNRI in the PNI < 48 or PNI ≥ 48 subgroup.

**Figure 7 cancers-13-05852-f007:**
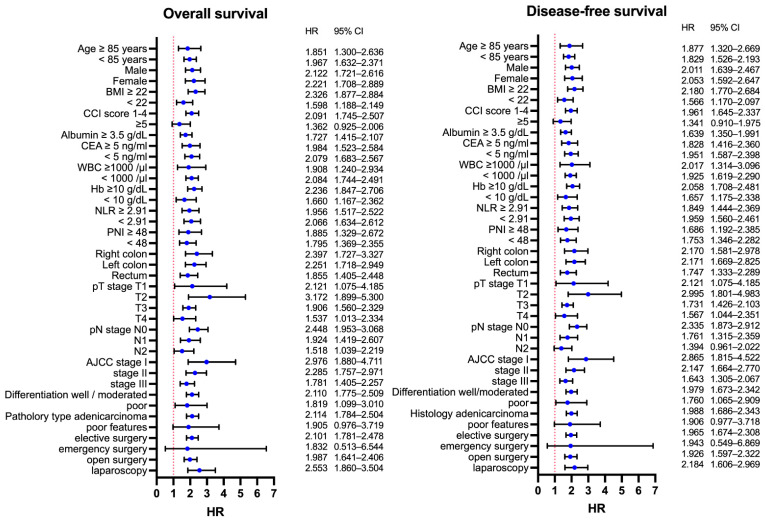
Summary of the subgroup survival analysis according to GNRI in each clinicopathological subgroup.

**Table 1 cancers-13-05852-t001:** Demographic characteristics of 1206 patients undergoing radical resection for stage I–III CRC.

Variables	All Patients (*n* = 1206)
Age	80.45 ± 4.42
Sex (male/female)	673/573 (55.8%/44.2%)
BMI	23.45 ± 3.70
Charlson comorbidity index score	
Mild (1–2)	423 (35.1)
Moderate (3–4)	627 (52.0)
Severe (≥5)	156 (12.9)
Albumin (g/dL)	3.82 ± 0.52
CEA (ng/mL)	11.76 ± 57.86
Operation type (elective/emergent)	1184/22
Operative method (open/laparoscopy)	848/358
Tumor location	
Right colon	351 (29.1)
Left colon	462 (38.2)
Rectum	393 (32.5)
GNRI	96.92 ± 8.82
Tumor invasion	
T1	84 (7)
T2	153 (12.7)
T3	807 (66.9)
T4	161 (13.3)
Lymph node metastasis	
N0	713 (59.1)
N1	332 (27.5)
N2	160 (13.3)
Stage	
Stage I	187 (15.5)
Stage II	527 (43.7)
Stage III	492 (40.8)
Follow up time (months)	63.01 ± 35.79
All postoperative complications	220 (18.2)
CD grade ≥ 2 complications	145 (12.0)

GNRI: Geriatric nutritional risk index; CD grade: Clavien-Dindo grade.

**Table 2 cancers-13-05852-t002:** The association of GNRI and clinicopathological factors in 1206 patients with CRC undergoing curative resection.

Variables	GNRI < 98 (*n* = 544)	GNRI ≥ 98 (*n* = 662)	*p* Value
Age	81.58 ± 4.80	79.51 ± 3.84	<0.001
Sex			
Male	291 (53.5)	382 (57.7)	0.143
Female	253 (46.5)	280 (42.3)	
BMI	21.74 ± 3.57	24.85 ± 3.70	<0.001
CCI score			
1–4	439 (80.7)	611 (92.3)	<0.001
≥5	105 (19.3)	51 (7.7)	
Albumin			
<3.5 g/dL	264 (48.5)	0 (0)	<0.001
≥3.5 g/dL	280 (51.5)	662 (100)	
CEA			
<5 ng/mL	319 (60.0)	462 (69.9)	<0.001
≥5 ng/mL	213 (40.0)	199 (30.1)	
WBC			
<1000/µL	435 (80)	599 (90.5)	<0.001
≥1000/µL	109 (20)	63 (9.5)	
Hb			
<10 g/dL	216 (39.7)	94 (14.2)	<0.001
≥10 g/dL	328 (60.3)	568 (85.8)	
NLR			
<2.91	241 (46.1)	421 (68.8)	<0.001
≥2.91	282 (53.9)	191 (31.2)	
PNI			
<48	454 (86.8)	145 (23.5)	<0.001
≥48	69 (13.2)	472 (76.5)	
Tumor location			
Right colon	201 (36.9)	150 (22.7)	<0.001
Left colon	200 (36.8)	262 (39.6)	
Rectum	143 (26.3)	250 (37.8)	
Neoadjuvant treatment *			
Yes	26 (18.2)	45 (18)	0.964
No	117 (81.8)	205 (82)	
Tumor invasion			
T1	24 (4.4)	60 (9.1)	<0.001
T2	50 (9.2)	103 (15.6)	
T3	362 (66.7)	445 (67.2)	
T4	107 (19.7)	54 (8.2)	
Lymph node metastasis			
N0	315 (58.0)	398 (60.1)	0.575
N1	150 (27.6)	182 (27.5)	
N2	78 (14.4)	81 (12.4)	
Stage			
Stage I	56 (10.3)	131 (19.8)	<0.001
Stage II	260 (47.8)	267 (40.3)	
Stage III	228 (41.9)	264 (40.8)	
Histological grade			
Well/moderately differentiated	473 (86.9)	615 (92.9)	0.001
Poorly differentiated	71 (13.1)	47 (7.1)	
Histological type			
Adenocarcinoma	489 (90.7)	638 (96.4)	<0.001
Poor features **	50 (9.3)	24 (3.6)	
Operative type			
Elective	528 (97.1)	656 (99.1)	0.009
Emergency	16 (2.9)	6 (0.9)	
Operative method			
Laparotomy	420 (77.2)	428 (64.7)	<0.001
Laparoscopy	124 (22.8)	234 (35.3)	

* For rectal cancer patient only; ** Poor features: mucinous/signet ring cell/undifferentiated adenocarcinoma.

**Table 3 cancers-13-05852-t003:** The association of GNRI and postoperative complications in 1206 patients with CRC undergoing curative resection.

Variables	GNRI < 98 (*n* = 544)	GNRI ≥ 98 (*n* = 662)	*p* Value
All complications	136 (25)	84 (12.7)	<0.001
Surgical site infection	40 (7.4)	23 (3.5)	0.003
Ileus	29 (5.3)	21 (3.2)	0.061
Leakage	7 (1.3)	12 (1.8)	0.465
Wound dehiscence	6 (1.1)	1 (0.2)	0.051 *
IAI	12 (2.2)	6 (0.9)	0.064
Pneumonia	37 (6.8)	7 (1.1)	<0.001
Urinary tract infection	18 (3.8)	8 (1.2)	0.012
Cardiovascular events	7 (1.3)	4 (0.6)	0.238 *

CD grade ≥ 2	100 (18.4)	45 (6.8)	<0.001
Mortality	16 (2.9)	2 (0.3)	<0.001
Postoperative nutrition intervention			
Total	19 (14)	5 (6)	0.178
Partial	17 (12.5)	12 (14.3)	

* Fisher’s exact test; IAI: intra-abdominal infection.

**Table 4 cancers-13-05852-t004:** Univariate and multivariate logistic regression analyses of the risk factors associated with CD grade ≥ 2 complications in patients with CRC undergoing a curative resection.

Variables	Univariate			Multivariate		
	HR	95% CI	*p* Value	HR	95% CI	*p* Value
Age (≥85/<85)	1.771	1.178–2.661	0.006	1.362	0.884–2.098	0.161
Sex (male/female)	1.177	0.827–1.675	0.365			
BMI (<22/≥22)	1.345	0.945–1.914	0.1			
CCI scores						
1–4	1			1		
≥5	2.358	1.537–3.617	<0.001	1.601	1.004–2.554	0.048
Albumin (<3.5/≥3.5)	2.767	1.930–3.988	<0.001	1.43	0.890–2.295	0.139
WBC count (≥1000)	1.605	1.030–2.500	0.037	1.011	0.607–1.685	0.966
Hb (<10)	1.207	0.821–1.773	0.339			
CEA (≥5)	1.047	0.725–1.513	0.805			
PNI (<48)	2.58	1.746–3.811	<0.001	1.184	0.677–2.070	0.554
NLR (≥2.91)	1.911	1.337–2.730	<0.001	1.275	0.832–1.955	0.264
GNRI (<98)	3.088	2.128–4.481	<0.001	1.774	1.035–3.040	0.037
Tumor location						
Left/right colon	0.715	0.466–1.097	0.125			
Rectum/right colon	0.92	0.602–1.408	0.701			
Tumor invasion						
T1	1					
T2	0.8	0.365–1.753	0.577			
T3	0.725	0.378–1.389	0.332			
T4	1.263	0.606–2.633	0.533			
LN metastasis						
N0	1					
N1	0.942	0.628–1.414	0.774			
N2	1.101	0.661–1.837	0.711			
Stage						
Stage I	1					
Stage II	0.848	0.515–1.397	0.518			
Stage III	0.883	0.535–1.458	0.627			
Histological grade						
Well/moderately differentiated	1					
Poorly differentiated	1.262	0.731–2.179	0.403			
Histological type						
Adenocarcinoma	1					
Poor features *	1.606	0.859–3.003	0.138			
Operative type (emergency)	3.537	1.417–8.828	0.007	2.177	0.821–5.773	0.118
Operative method (open)	1.123	0.763–1.655	0.556			
Neoadjuvant treatment (Yes/No) **	1.156	0.548–2.437	0.704			

** For rectal cancer patient only; * Poor features: mucinous/signet ring cell/undifferentiated adenocarcinoma.

**Table 5 cancers-13-05852-t005:** Univariate and multivariate analyses of prognostic factors for OS in patients with CRC undergoing a curative resection.

Variables	Univariate			Multivariate		
	HR	95% CI	*p* Value	HR	95% CI	*p* Value
Age (≥85/<85)	2.34	1.940–2.822	<0.001	2.046	1.673–2.502	<0.001
Sex (male/female)	1.271	1.078–1.499	0.004	1.332	1.118–1.587	0.001
BMI (<22/≥22)	1.485	1.261–1.749	<0.001	1.236	1.026–1.489	0.026
CCI scores						
1–4	1			1		
≥5	2.517	2.055–3.084	<0.001	2.081	1.658–2.612	<0.001
Albumin (<3.5/≥3.5)	2.237	1.875–2.668	<0.001	1.275	1.009–1.611	0.042
WBC count (≥1000)	1.495	1.208–1.849	<0.001	1.005	0.784–1.287	0.971
Hb (<10)	1.331	1.113–1.593	0.002	0.887	0.724–1.086	0.246
CEA (≥5)	1.461	1.237–1.724	<0.001	1.162	0.969–1.395	0.105
PNI (<48)	1.938	1.631–2.302	<0.001	1.145	0.889–1.474	0.295
NLR (≥2.91)	1.688	1.429–1.995	<0.001	1.208	0.989–1.476	0.064
GNRI (<98)	2.115	1.796–2.490	<0.001	1.329	1.026–1.722	0.031
Tumor location						
Left colon/right colon	0.804	0.707–1.055	0.152			
Rectum/right colon	0.971	0.792–1.189	0.773			
Tumor invasion						
T1	1			1		
T2	0.929	0.612–1.412	0.73	0.924	0.598–1.428	0.723
T3	1.221	0.864–1.726	0.258	0.904	0.621–1.316	0.904
T4	2.453	1.675–3.592	<0.001	1.473	0.959–2.263	0.077
Lymph node metastasis						
N0	1			1		
N1	1.264	1.048–1.524	0.014	1.27	1.036–1.556	0.021
N2	2.053	1.649–2.557	<0.001	2.061	1.623–2.618	<0.001
Stage						
Stage I	1					
Stage II	1.178	0.908–1.529	0.218			
Stage III	1.685	1.303–2.177	<0.001			
Histological grade						
Well/moderately differentiated	1			1		
Poorly differentiated	1.503	1.171–1.929	0.001	1.127	0.812–1.564	0.475
Histological type						
Adenocarcinoma	1			1		
Poor features *	1.537	1.137–2.077	0.005	1.127	0.766–1.656	0.544
Operative type (emergency)	1.951	1.168–3.259	0.011	1.003	0.539–1.868	0.991
Operative method (open)	1.093	0.909–1.315	0.344			

* Poor features: mucinous/signet ring cell/undifferentiated adenocarcinoma.

**Table 6 cancers-13-05852-t006:** Univariate and multivariate analysis of prognostic factors for DFS in patients with CRC undergoing a curative resection.

Variables	Univariate			Multivariate		
	HR	95% CI	*p* Value	HR	95% CI	*p* Value
Age (≥85/<85)	2.145	1.783–2.579	<0.001	1.9	1.560–2.316	<0.001
Sex (male/female)	1.279	1.090–1.502	0.003	1.341	1.130–1.591	0.001
BMI (<22/≥22)	1.394	1.188–1.636	<0.001	1.186	0.988–1.423	0.067
CCI scores						
1–4	1			1		
≥5	2.262	1.849–2.768	<0.001	1.938	1.547–2.429	<0.001
Albumin (<3.5/≥3.5)	2.091	1.759–2.486	<0.001	1.201	0.954–1.512	0.118
WBC count (≥1000)	1.427	1.157–1.759	0.001	0.937	0.734–1.195	0.599
Hb (<10)	1.309	1.099–1.559	0.002	0.88	0.722–1.073	0.207
CEA (≥5)	1.554	1.323–1.827	<0.001	1.259	1.054–1.503	0.011
PNI (<48)	1.866	1.578–2.207	<0.001	1.162	0.911–1.484	0.227
NLR (≥2.91)	1.633	1.387–1.922	<0.001	1.194	0.982–1.452	0.075
GNRI (<98)	1.985	1.694–2.327	<0.001	1.312	1.021–1.685	0.034
Tumor location						
Left colon/right colon	0.885	0.728–1.076	0.222			
Rectum/right colon	0.998	0.819–1.217	0.987			
Tumor invasion						
T1	1			1		
T2	0.951	0.627–1.443	0.814	0.94	0.609–1.450	0.78
T3	1.364	0.966–1.927	0.078	1.01	0.695–1.469	0.957
T4	2.734	1.871–3.994	<0.001	1.675	1.096–2.560	0.017
Lymph node metastasis						
N0	1			1		
N1	1.337	1.115–1.603	0.002	1.345	1.106–1.637	0.003
N2	2.222	1.792–2.756	<0.001	2.212	1.748–2.799	<0.001
Stage						
Stage I	1					
Stage II	1.264	0.977–1.636	0.075			
Stage III	1.883	1.461–2.426	<0.001			
Histological grade						
Well/moderately differentiated	1					
Poorly differentiated	1.414	1.105–1.811	0.006	1.018	0.739–1.402	0.913
Histological type						
Adenocarcinoma	1					
Poor features *	1.407	1.042–1.901	0.026	1.057	0.725–1.541	0.773
Operative type (emergency)	2.074	1.262–3.408	0.004	1.035	0.572–1.874	0.909
Operative method (open)	1.1	0.919–1.315	0.298			

* Poor features: mucinous/signet ring cell/undifferentiated adenocarcinoma.

## Data Availability

The datasets generated and analyzed during the current study are available from the corresponding author upon reasonable request.

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
