# Peer review of "The Clinical Utility of the Geriatric Nutritional Risk Index in Predicting Postoperative Complications and Long-Term Survival in Elderly Patients with Colorectal Cancer after Curative Surgery"

_cancers, 2021, doi:10.3390/cancers13225852_

Round 1

Reviewer 1 Report

At present, the nutritional evaluation of the surgical patient is increasingly important as it conditions complications and survival. The authors present a nice paper on this topic.
However, I wanted to make the following observations:
1. The authors themselves mark 75 years as age against 65 years (in which I agree for the concept "geriatric"). However, it would be interesting if they included the 65-75 age group separately in their analysis to see if there were differences or not.
2. The inclusion and exclusion criteria are opposite (so you are repeating the same thing in the text).
3. One of the criteria is the lack of data (since it is a retrospective study). How many went?
4. One aspect that could create confusion is the preoperative nutritional status. Was screening done with any scale? Must, NRS, etc etc?
5. Was any nutritional intervention made? To what and how many patients? conditioned the results?
6. It is increasingly proven that it is not only BMI that conditions nutritional status. We can have normal BMI in malnourished patients, high BMI (overweight) and be malnourished and they are patients clearly with a tendency to complicate, and low BMI with normal nutrition. I am seriously concerned that there is a bias and looking at just Alb and BMI is not enough. This should be discussed.
7. Preop factors should be added such as neoadjuvant rectal cancer that influences the results.
8. The biases in table 2 in both groups should be discussed.

Author Response

We deeply appreciate these kindly and creative feedback and comment from distinguished reviewer. Below are our response point-by-point. Please see the attachment.

Reviewer 2 Report

I found your paper very interesting and with a suitable objective in the field of CRC elderly cancer patients.
However, I have some doubt about the opportunity to include in the results albumin and BMI as factors significantly related to a lower GNRI: in fact they are the parameters included in the tool that you used, so in my opinion is pleonastic to cite that.
Moreover, a little observation in the "summary": I think that "body height" is a very rarely used term to define height.
Generally speaking, I think that the study is correctly presented and potentially useful in clinical practice.

Author Response

(The authors gave the same response as above.)

Round 2

Reviewer 1 Report

changues ok